# Nuclear genetic regulation of the human mitochondrial transcriptome

**Aminah T Ali[1], Lena Boehme[1], Guillermo Carbajosa[1], Vlad C Seitan[1], Kerrin S Small[2], Alan Hodgkinson[1]\***

[1]Department of Medical and Molecular Genetics, School of Basic and Medical Biosciences, King's College London, London, United Kingdom; [2]Department of Twin Research and Genetic Epidemiology, School of Life Course Sciences, King's College London, London, United Kingdom

**Abstract** Mitochondria play important roles in cellular processes and disease, yet little is known about how the transcriptional regime of the mitochondrial genome varies across individuals and tissues. By analyzing >11,000 RNA-sequencing libraries across 36 tissue/cell types, we find considerable variation in mitochondrial-encoded gene expression along the mitochondrial transcriptome, across tissues and between individuals, highlighting the importance of cell-type specific and post-transcriptional processes in shaping mitochondrial-encoded RNA levels. Using whole-genome genetic data we identify 64 nuclear loci associated with expression levels of 14 genes encoded in the mitochondrial genome, including missense variants within genes involved in mitochondrial function (*TBRG4*, *MTPAP* and *LONP1*), implicating genetic mechanisms that act in *trans* across the two genomes. We replicate ~21% of associations with independent tissue-matched datasets and find genetic variants linked to these nuclear loci that are associated with cardio-metabolic phenotypes and Vitiligo, supporting a potential role for variable mitochondrial-encoded gene expression in complex disease.
DOI: https://doi.org/10.7554/eLife.41927.001

**\*For correspondence:**
alan.hodgkinson@kcl.ac.uk

**Competing interests:** The authors declare that no competing interests exist.

## Introduction

Mitochondria are involved in a wide range of fundamental cellular processes, including cellular energy production, thermogenesis, lipid biosynthesis and cell death, and mutations in both nuclear and mitochondrial DNA (mtDNA) encoded genes have been linked to an array of different diseases (*Taylor and Turnbull, 2005*; *He et al., 2010*; *Nunnari and Suomalainen, 2012*; *Hudson et al., 2014*; *Idaghdour and Hodgkinson, 2017*). Most of the genes encoded in the mitochondrial genome are transcribed as one strand of RNA, and post-transcriptional processes are therefore particularly important for gene regulation. After transcription, poly-cistronic mitochondrial RNA is processed under the 'punctuation model' whereby transfer RNAs (tRNAs) that intersperse protein-coding regions are recognized for cleavage and the release of gene products (*Ojala et al., 1981*; *Sanchez et al., 2011*). Various processes including RNA modifications (*Helm et al., 1998*; *Helm et al., 1999*; *Agris et al., 2007*), further cleavage events (*Mercer et al., 2011*; *Rackham et al., 2012*), RNA degradation (*Sasarman et al., 2010*; *Rackham et al., 2011*) and translation rates then ultimately determine the levels of mitochondrial proteins available for utilization in the electron transport chain. Across tissues, different cell types have specific physiological requirements and thus variable energy demands. In mammals it has been shown that mitochondrial DNA replication (*Herbers et al., 2019*) and segregation (*Jokinen et al., 2010*), mitochondrial DNA copy number (*Wachsmuth et al., 2016*) and the abundance of nuclear-encoded mitochondrial proteins (*Mootha et al., 2003*) vary across cell types, perhaps as a way to match local energy requirements, however it is unclear whether regulation of the mitochondrial transcriptome varies across tissues.

**eLife digest** Mitochondria are like the batteries of our cells; they perform the essential task of turning nutrients into chemical energy. A cell relies on its mitochondria for its survival, but they are not completely under the cell's control. Mitochondria have their own DNA, separate from the cell's DNA which is stored in the nucleus. It contains a handful of genes, which carry the code for some of the important proteins needed for energy production.

These proteins are made in the mitochondria themselves, and their levels are tweaked to meet the cell's current energy needs. To do this, mitochondria make copies of their genes and feed these copies into their own protein-production machinery. By controlling the number of gene copies they make, mitochondria can control the amount of protein they produce. But the process has several steps. The copies come in the form of a DNA-like molecule called RNA and, at first, they contain several genes connected one after the other. To access each gene, the mitochondria need to cut them up. They then process the fragments, fine-tuning the number of copies of each gene. This process – called gene expression – happens in the mitochondria, but they cannot do it on their own; they need proteins that are coded within the DNA in the cell nucleus.

Genes in the cell nucleus can affect gene expression in the mitochondria, changing the cell's energy supply. Scientists do not yet know all of the genes involved, or how this might differ between different tissues or among different individuals. To find out, Ali et al. examined more than 11,000 records of RNA sequences from 36 different human cells and tissues, including blood, fat and skin. This revealed a large amount of variation in the expression of mitochondrial genes. The way the mitochondria processed their genes changed in different cells and in different people. To find out which genes in the nucleus were responsible for the differences in the mitochondria, the next step was to compare RNA levels from the mitochondria to the DNA sequences in the nucleus. This is because changes in the DNA sequence between different people – called genetic variants – can also affect how genes work, and how genes are expressed. This comparison revealed 64 genetic variants from DNA in the cell nucleus that are associated with the expression of genes in the mitochondria. Some of these had a known link to genetic variants involved in diseases like the skin condition vitiligo or high blood pressure.

So, although mitochondria contain their own DNA, they rely on genes from the cell nucleus to function. Changes to the genes in the nucleus can alter the way that the mitochondria process their own genetic code. Understanding how these two sets of genes interact could reveal how and why mitochondria go wrong. This could aid in future research into illnesses like heart disease and cancer.

DOI: https://doi.org/10.7554/eLife.41927.002

Understanding these processes is important, since many mitochondrial disorders are thought to be tissue specific (*Koppen et al., 2007*; *Hämäläinen et al., 2013*).

Although the mitochondrial genome is transcribed, processed and translated within the mitochondria, almost all of the proteins required for these processes are coded for in the nuclear genome. Previous work has shown that the expression of a large number nuclear genes correlates with mitochondrial encoded gene expression (*Mercer et al., 2011*; *Barshad et al., 2018*), pointing to strong links between the two genomes, yet there is still not a complete understanding of which nuclear genes are directly involved in regulating the mitochondrial genome and how this might vary in different tissues, as well as whether nuclear genetic variation drives variation in these processes across individuals. Despite the wide-ranging impact of mitochondrial dysfunction on health and disease, to our knowledge only a single mitochondria-focussed study has been carried out comparing nuclear genome-wide genetic variation with mitochondrial encoded gene expression, which analysed two sets of ~70 samples and was underpowered to detect genetic variation acting across two genomes (*Wang et al., 2014*). More recently, studies have shown links between mitochondrial genome mutations and nuclear gene expression, identifying 11 significant associations (*Kassam et al., 2016*), as well as associations between single nucleotide polymorphisms (SNPs) in mitochondrial RNA-binding proteins and haplogroup-specific mtDNA encoded gene expression patterns in LCLs (*Cohen et al., 2016*), providing good evidence for regulatory links between the two genomes. In general, genetic variation associated with the expression of distal genes (*trans*

expression quantitative trait loci (eQTLs)) has been more difficult to find due to the large statistical burden when comparing large numbers of variants and genes, and very few significant associations have been replicated in independent datasets (*Innocenti et al., 2011*; *Kirsten et al., 2015*; *GTEx Consortium et al., 2017*).

Here we aim to characterize variation in mitochondrial encoded gene expression across >11,000 RNA sequencing libraries for 36 different tissue/cell types. We also aim to identify genetic links between the mitochondrial and nuclear genomes through the detection of *trans*-genome eQTLs, not only to evidence occasions where genetic mechanisms act at long range across different genetic regions, but also to identify novel genes and genetic variation in the nuclear genome that are associated with fundamental processes taking place in human mitochondria.

## Results

To characterize levels of mitochondrial encoded RNA across a large number of individuals and tissues, we obtained raw RNA sequencing data for 13,261 samples from five independent sequencing projects, covering 36 different tissue/cell types, including multiple independent datasets obtained from whole blood, subcutaneous adipose, skin (not sun exposed) and lymphoblastic cell lines (LCLs). For each dataset, sequencing data were processed consistently using the same stringent mapping and filtering pipeline (see Materials and methods), removing poor quality samples at each stage, leaving a total of 11,371 high quality samples for comparison (*Figure 1A*), allowing us to focus on biological rather than technical variation. Following this, expression levels were quantified as the number of transcripts per million reads (TPM) per sample for 13 protein-coding genes and two ribosomal RNAs encoded in the mitochondrial genome.

### Variation in mitochondrial gene expression

Overall, despite their polycistronic origins, there is significant variation between mean expression levels of the 15 mitochondria-encoded genes within each dataset (one-way ANOVA, p<2e-16 in all cases), highlighting the influence of post-transcriptional events in generating variation in transcript abundance along the mitochondrial transcriptome in all tissues. On average across samples and datasets, *MTCO3* and *MTCO2* show the highest median expression levels and *MTRNR1* the lowest. Hierarchical clustering of log median expression values per dataset shows the consistency of the data, as the same tissue types from independent sequencing datasets generally tend to cluster together (*Figure 1A*). Whole blood, LCL and skin datasets group by tissue type, however subcutaneous adipose data do not; this may be a consequence of the large heterogeneity in cell type composition observed across these datasets (*Glastonbury et al., 2018*). High-energy tissues (for example heart and brain tissues) also tend to cluster together and appear to show similar patterns of mitochondrial encoded gene expression.

In general, the rank order of mitochondrial-encoded gene expression levels between tissues is broadly similar (spearman rank rho >0.5 for 894/903 pairwise comparisons of independent datasets) with genes that show high relative expression levels in one tissue tending to show high relative expression levels in others tissues, however there are gene specific patterns. Standardized median *MTRNR2* expression levels are highly variable, showing higher relative expression in whole blood and sub regions of the brain compared to other tissue types, whereas *MTND4L*, *MTND5* and *MTATP8* have low variance across tissue types and show relatively low standardized expression (*Figure 1B*). Across individuals within each tissue, mitochondria-encoded genes show similar variance to comparable nuclear genes; on average across genes and datasets, the coefficient of variation of mitochondrial encoded TPM values is higher than 443 of the top 1000 most highly expressed nuclear genes and distributions of coefficients of variation overlap (*Figure 1C*). However, there are differences across tissues; mitochondrial encoded genes in sub-regions of the brain generally show low variation in gene expression across individuals, and expression variance in whole blood is generally high. Collectively these results point to significant variation in the expression of genes along the mitochondrial genome, across tissues and across individuals.

### Nuclear control of mitochondrial gene expression

To identify nuclear genetic variation associated with mitochondrial encoded transcript abundance, we obtained genotyping data for the same samples for which we had RNA sequencing data and

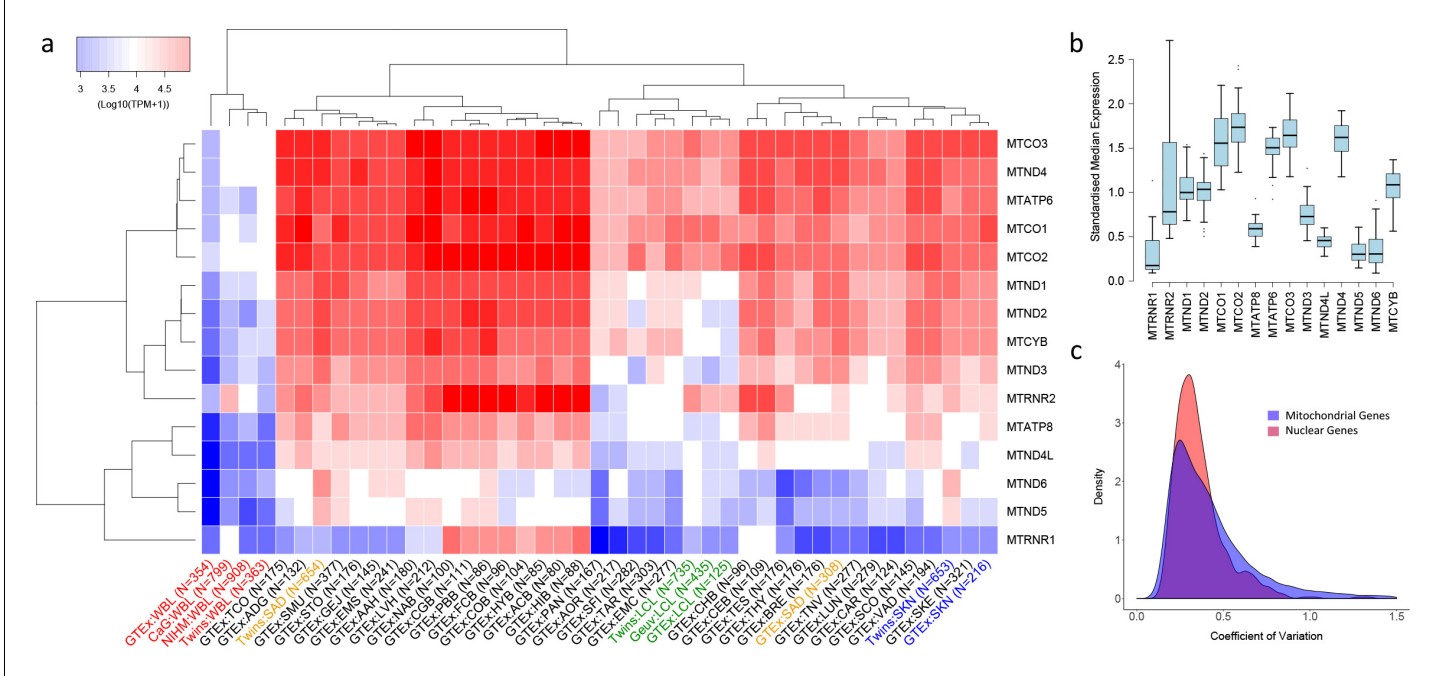

**Figure 1.** Variation in the expression of mitochondrial-encoded genes across datasets. (A) Hierarchical clustering of median expression levels per gene across all datasets where WBL = Whole Blood, SAD = Subcutaneous Adipose, LCL = Lymphoblastoid cell lines, SKN = Non sun exposed skin, SKE = Sun exposed skin, VAD = Visceral omentum adipose, ADG = Adrenal gland, AOR = Aorta, CAR = Coronary artery, TAR = Tibial artery, ACB = Anterior cingulate cortex (BA24) (Brain), CGB = Caudate basal ganglia (Brain), CHB = Cerebellar Hemisphere (Brain), CEB = Cerebellum (Brain), COB = Cortex (Brain), FCB = Frontal cortex (BA9) (Brain), HIB = Hippocampus (Brain), HYB = Hypothalamus (Brain), NAB = Nucleus accumbens (basal ganglia) (Brain), PBB = Putamen basal ganglia (Brain), BRE = Breast mammary tissue, SCO = Sigmoid colon, TCO = Transverse colon, GEJ = Gastroesophageal junction, EMC = Esophagus mucosa, EMS = Esophagus Muscularis, AAH = Atrial appendage (Heart), LVH = Left ventricle (Heart), LUN = Lung, SMU = Skeletal muscle, TNV = Tibial Nerve, PAN = Pancreas, SFI = Transformed fibroblasts, STO = Stomach, TES = Testes and THY = Thyroid, Multi-dataset tissues on the x-axis are shown in red (whole blood), orange (subcutaneous adipose), green (lymphoblastoid cell lines) and blue (non-sun exposed skin). (B) Standardized expression levels of each mitochondrial-encoded gene across all independent datasets, (C) Coefficient of variation across individuals for the expression levels of mitochondrial encoded genes and the top 1000 most highly expressed nuclear genes in all datasets. Range of coefficient of variation is restricted to between 0 and 1.5 as this contains the majority of the data.

DOI: https://doi.org/10.7554/eLife.41927.003

then performed per tissue and dataset association analyses between nuclear genetic variants (with MAF >5%) and the expression levels of fifteen mitochondrial encoded genes within a linear model, controlling for ancestry, sex, batch (where applicable) and probabilistic estimation of expression residuals (PEER factors) (*Stegle et al., 2010*) obtained from RNA sequencing data. For whole blood, subcutaneous adipose, non-sun exposed skin and LCLs where we had multiple independent datasets, we defined discovery and replication datasets.

Across all tissues, we identify a total of 64 *trans*-genome eQTLs (unique peak genetic variant-gene expression pairs) for mitochondrial encoded gene expression at FDR 5% (range of FDR corrected p-values: $0.046 - 8 \times 10^{-26}$, *Supplementary file 1*, example association shown in *Figure 2A and C*). For each significant association, we also calculate point-wise empirical P-values (as well as gene-level and tissue-level family-wise error rates) via permutation analysis, and find that these closely match raw P-values (see Materials and methods and *supplementary file 1*). In total, fourteen out of the fifteen mitochondrial encoded genes have at least one nuclear genetic variant associated with its expression; *MTATP8* shows no significant associations, *MTND1* has the most with seven independent associations. We also observe five instances where a peak nuclear variant is associated with the expression of multiple mitochondrial-encoded genes within a tissue, perhaps indicating a shared influence on mitochondria RNA processing. However, mitochondrial encoded genes associated with the same genetic variant are no more likely to be located closer to each other along the mitochondrial genome than random (p=0.29, bootstrapping versus same number of random chosen

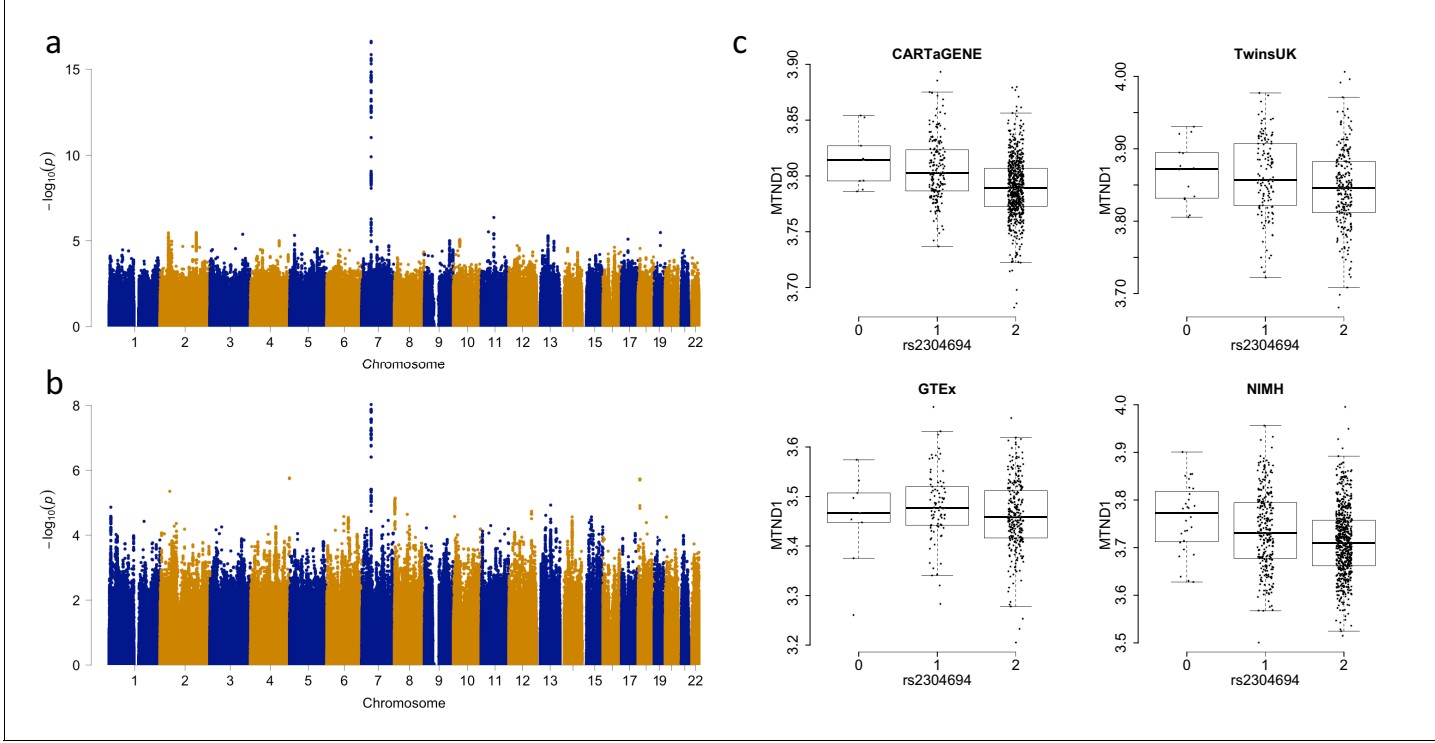

**Figure 2.** Associations between the expression of MTND1 and rs2304694 in whole blood data. (A) Genome-wide association analysis for the expression of *MTND1* in whole blood data from the discovery datasets (meta-analysis of CARTaGENE, TwinsUK and GTEx data), (B) Genome-wide association analysis for the expression of *MTND1* in whole blood data from the replication dataset (NIMH data), (C) Expression of *MTND1* ($Log_{10}(TPM +1)$) versus non-reference allele frequency of rs2304694 in the four independent whole blood datasets.

DOI: https://doi.org/10.7554/eLife.41927.004

The following figure supplement is available for figure 2:

**Figure supplement 1.** QQ plots for associations between nuclear genetic variants and mitochondrial gene expression for discovery associations that replicate at the nominal 5% level.

DOI: https://doi.org/10.7554/eLife.41927.005

genes). For the 49 unique peak genetic variants remaining after removing duplicate variants with multiple associations, four are missense mutations, 32 intronic, 12 intergenic and one falls in a 3' UTR region.

To ensure that *trans*-genome eQTLs are not driven by alignment errors that are a consequence of sequence similarity between the nuclear and the mitochondrial genomes, we tested for the presence of nuclear mitochondrial DNA segments (NUMTs) in the regions surrounding each peak nuclear genetic variant. NUMTs are mitochondrial DNA sequences that have transposed into the nuclear genome over evolutionary time scales, and as such often retain moderate to high sequence similarity with the mitochondrial genome. For the 64 trans-genome eQTLs, we find only two occurrences where at least 50 bp (the smallest read length in our analysis) of the mitochondrial encoded gene is present within a NUMT that is within 1 MB of the corresponding peak nuclear genetic variant, and we observe ~4 and~15 mismatches per 100 bp in these sequences compared to the corresponding mitochondrial encoded sequence. Additionally, for each peak nuclear genetic variant that is associated with the expression of a mitochondrial-encoded gene, we also tested whether any 50 bp segment of the mitochondrial-encoded gene also mapped to a nuclear gene (following the approach defined in *Saha and Battle, 2018*) that has its transcription start site within 1 MB of the corresponding peak nuclear variant; we find no such occurrences. As such, alignment errors are unlikely to be driving the detection of *trans*-genome eQTLs for mitochondrial encoded gene expression.

RNA levels of mitochondrial-encoded genes are likely driven by a number of features including mitochondrial copy number, polycistronic transcription rates and post-transcriptional events. Although all of these processes are important in a biological context, after detecting initial

**Table 1.** Associations where a suggestive causal nuclear gene is implicated.

'Missense mutation' denotes that the nuclear genetic variant associated with the expression of a mitochondrial-encoded gene is a missense mutation, 'Mediation (Mitochondrial Gene)' denotes that the expression of a nearby nuclear gene known to play a role in mitochondrial processes explains a significant proportion of the association between a nuclear genetic variant and the expression level of a mitochondrial encoded gene, and 'Mediation (other nuclear gene)' denotes a similar result whereby the nuclear gene identified is thought to have no known role in mitochondrial processes (see Materials and methods).

| Tissue | Peak SNP | MT gene | Missense mutation | Mediation (Mitochondrial Gene) | Mediation (other nuclear gene) |
|---|---|---|---|---|---|
| Whole Blood | rs7558127 | MTND6 | NA | PNPT1 | NA |
| Whole Blood | rs6973982 | MTCO2 | NA | TBRG4 | NA |
| Whole Blood | rs11085147 | MTCO2 | LONP1 | NA | NA |
| Whole Blood | rs2304693 | MTCYB | TBRG4 | NA | NA |
| Whole Blood | rs74025341 | MTCYB | NA | NA | SLC7A6OS,ZFP90 |
| Whole Blood | rs7158706 | MTND2 | NA | NA | PPP2R3C |
| Whole Blood | rs10172506 | MTND5 | NA | PNPT1 | NA |
| Whole Blood | rs74863981 | MTCO1 | NA | NA | UBOX5,TGM3,LZTS3 |
| Whole Blood | rs76125482 | MTND3 | NA | FASTKD1 | NA |
| Whole Blood | rs6973982 | MTND4 | NA | NA | RP4-647J21.1,CCM2 |
| Whole Blood | rs11008009 | MTND4 | NA | MTPAP | NA |
| Whole Blood | rs2304694 | MTND1 | TBRG4 | NA | NA |
| Whole Blood | rs1692120 | MTND1 | NA | NA | MYRF |
| Whole Blood | rs6973982 | MTATP6 | NA | NA | CCM2 |
| Whole Blood | rs589809 | MTATP6 | NA | NA | FLT1 |
| Whole Blood | rs375640557 | MTCO3 | NA | NA | CCDC104 |
| Whole Blood | rs6973982 | MTCO3 | NA | TBRG4 | NA |
| Whole Blood | rs10165864 | MTRNR2 | NA | PNPT1 | NA |
| Whole Blood | rs66892251 | MTRNR2 | NA | MTPAP | NA |
| Whole Blood | rs61988269 | MTRNR1 | NA | MRPP3 | NA |
| Subcutaneous Adipose | rs2304694 | MTND6 | TBRG4 | NA | NA |
| Subcutaneous Adipose | rs2304694 | MTND5 | TBRG4 | NA | NA |
| Subcutaneous Adipose | rs2304694 | MTND1 | TBRG4 | NA | NA |
| Subcutaneous Adipose | rs12579998 | MTND1 | NA | MRPS35 | NA |
| Subcutaneous Adipose | rs2304693 | MTCO3 | TBRG4 | NA | NA |
| Skin (Not sun exposed) | rs2304693 | MTCO2 | TBRG4 | NA | NA |
| Skin (Not sun exposed) | rs2304693 | MTCO3 | TBRG4 | NA | NA |
| LCLs | rs7559561 | MTCO2 | NA | LRPPRC | NA |
| LCLs | rs2304694 | MTCO2 | TBRG4 | NA | NA |
| LCLs | rs1047991 | MTND3 | MTPAP | NA | NA |
| LCLs | rs2304694 | MTND4 | TBRG4 | NA | NA |
| LCLs | rs10205130 | MTND1 | NA | LRPPRC | NA |
| LCLs | rs35739334 | MTND1 | NA | TBRG4 | NA |
| LCLs | rs2304694 | MTCO3 | TBRG4 | NA | NA |
| LCLs | rs2304694 | MTRNR2 | TBRG4 | NA | NA |
| LCLs | rs2304694 | MTND4L | TBRG4 | NA | NA |

DOI: https://doi.org/10.7554/eLife.41927.006

associations we focussed on the effects of post-transcriptional processing in driving variation in mitochondrial encoded gene expression. To do this, we controlled for variable mitochondrial copy

number and polycistronic transcription rate by recalculating TPM values for each mitochondrial gene and sample using the number of mitochondrial reads rather than the total RNA sequencing library size. Repeating association analyses as before, 63/64 associations remain significant at FDR 5% (*Supplementary file 2*). Since mitochondrial encoded gene expression values are represented as a proportion of the total reads mapping to the mitochondrial genome in this analysis, this suggests that post-transcriptional processes play a significant role in these associations.

To identify whether genetic associations are tissue specific, for the 64 significant associations we tested whether the same peak variant-gene pair was significant with the same direction of effect in each of the other tissue types (at p<0.05, corrected for the number of variants and the number of tissues, we used the nearest variant in LD ($r^2$ >0.8) if the same variant was not present, or the nearest variant with $r^2$ >0.5 otherwise). In total, 22 of the 64 associations are significant in more than one tissue, with 8 of the associations being observed in at least three other tissue types (*Supplementary file 3*, *Supplementary file 6*). Lowering the p-value threshold to 5% with the same direction of effect, only 12 associations are not replicated outside of the tissue they were originally detected in, and 19 associations are significant across 10 or more tissue types. Although sample sizes and detection criteria may influence our ability to detect all associations, these results indicate that a large number of associations between the nuclear and mitochondrial genomes may be operating via general mechanisms that occur across multiple tissue types.

## Functional characterization

In order to elucidate the potential biological mechanisms influencing mitochondrial processes, we attempted to identify the nuclear gene of action through which each nuclear genetic variant is associated with mitochondrial encoded gene expression. For missense variants, we assume a direct influence on the gene in which they are located and thus identify three nuclear genes associated with mitochondrial encoded gene expression (*Table 1*), all of which have a known role in mitochondrial processes. *TBRG4* localizes to the mitochondria to modulate energy balance (particularly under stress) and plays a role in processing mitochondrial RNA (*Boehm et al., 2017*), *MTPAP* synthesizes the 3' poly(A) tail of mitochondrial transcripts, and *LONP1* mediates the degradation of mis-folded or damaged polypeptides in the mitochondrial matrix. There is evidence that all three proteins are targeted to the mitochondria, and mass spectrometry experiments have identified the presence of these proteins in mitochondria (*Smith and Robinson, 2016*).

For genetic variants in non-coding regions (from 49 unique associations), we first annotated variants using chromatin state predictions obtained from 128 cell types within the Roadmap Epigenetic project (*Kundaje et al., 2015*). Using tissue matched data (information available for 44 of the 49 non-coding variants), we find that none of the nuclear genetic variants associated with mitochondrial encoded gene expression fall in enhancer regions, which is not different to that expected by chance (p=0.676 using randomly selected variants matched for MAF, distance to nearest transcription start site and annotation). Under the assumption that associations between nuclear genetic variants and mitochondrial encoded gene expression occur ubiquitously across the body, we tested for the presence of peak variants in enhancer regions in any cell type. In total, 24 variants fall in enhancer regions, which again is not significantly different from that expected by chance (p=0.691, using randomly selected variants as before).

To test more directly if each nuclear non-coding genetic variant potentially acts upon mitochondrial-encoded gene expression through a nearby nuclear gene, we perform mediation analysis (requiring an association between the peak nuclear genetic variant and the expression of a nearby nuclear-encoded gene, and then significant mediation of the initial association via bootstrapping, requiring an average causal mediation effect with p<0.05 after FDR correction). Considering only nuclear genes known to play a role in mitochondrial processes first, we identify seven genes whose expression accounts for a significant component of the relationship between the nearby nuclear genetic variant and the expression of the associated mitochondrial-encoded gene (*Table 1*). These include *TBRG4* and *MTPAP* (described above), as well as *MRPP3*, which is known to form part of a complex that cleaves and processes the 5' end of mitochondrial transfer RNAs (*Holzmann et al., 2008*); *LRPPRC*, which is thought to play a role in the stability and transcriptional regulation of mitochondrial RNA (*Xu et al., 2004*); *MRPS35*, which is a mitochondrial ribosomal protein; *PNPT1*, which is an RNA binding protein that plays a role in numerous RNA metabolic processes and the import of RNA into the mitochondria; and *FASTKD1*, which is an RNA binding protein that regulates the

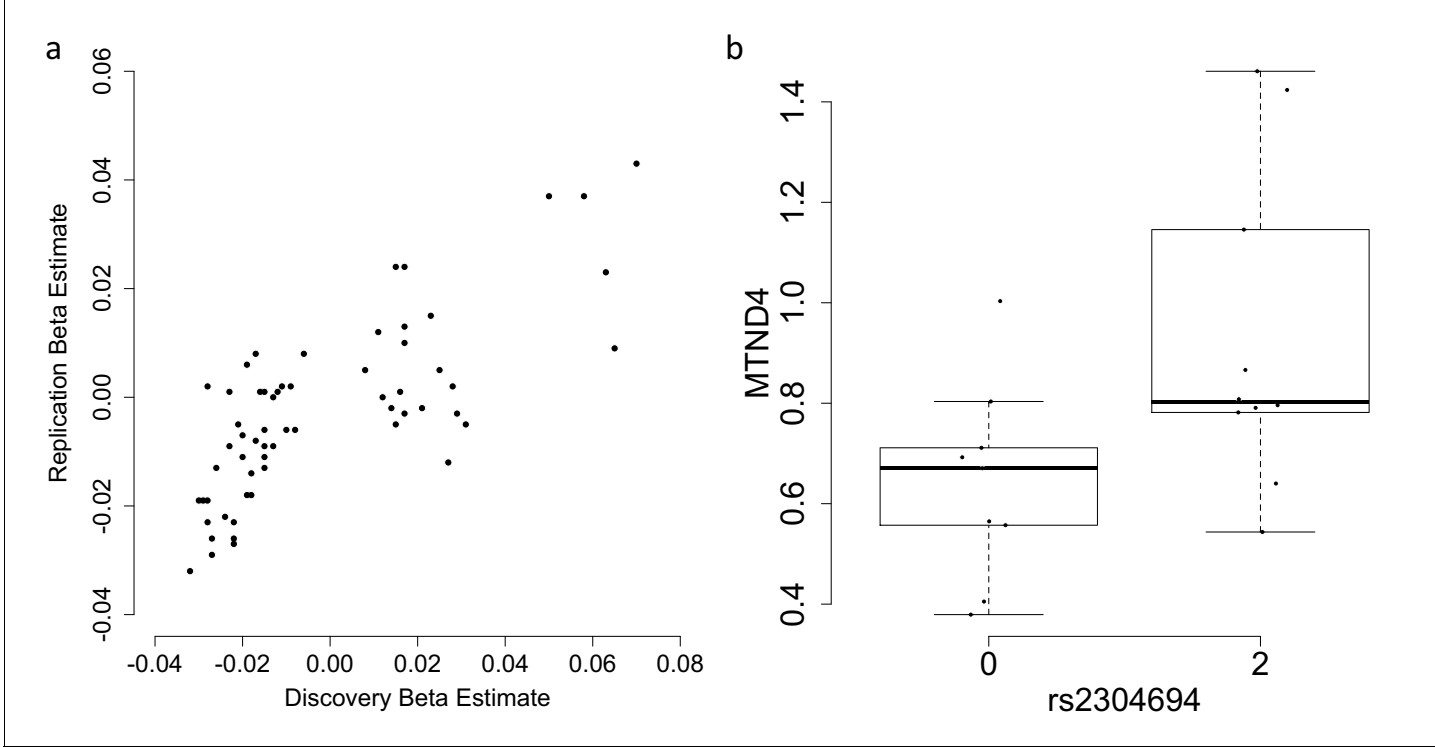

**Figure 3.** Replication and validation of significant associations between nuclear genetic variants and the expression of mitochondria-encoded genes. (A) Discovery versus replication beta estimates for significant associations between nuclear genetic variation and mitochondrial gene expression detected in discovery data at FDR 5%, (B) Validation of the association between rs2304694 and the expression of *MTND4* using quantitative PCR in LCLs. *MTND4* mRNA expression levels are normalised to GAPDH (theoretical quantities).
DOI: https://doi.org/10.7554/eLife.41927.007

energy balance of mitochondria under stress. Five out of the seven proteins (*PNPT1*, *TBRG4*, *MTPAP*, *MRPP3* and *LRPPRC*) contain RNA binding domains (**Wolf and Mootha, 2014**), and as such it is possible that they bind directly to mitochondrial RNA.

For the remaining non-coding peak genetic variants (from 36 unique associations), we tested whether any nearby nuclear genes (not yet implicated in mitochondrial processes) significantly mediated the expression of a mitochondria-encoded gene (as above). Using this approach, we identify eleven candidate genes that may play a previously unknown role in influencing mitochondrial gene expression (**Table 1**). In general, these genes are not predicted to contain mitochondria targeting sequences, although *SLC7A6OS* and *TGM3* show partial evidence of being targeted to mitochondria in some databases (*SLC7A6OS* prediction score of 1 in IPSort and both genes have a score >0.6 in TargetP (**Smith and Robinson, 2016**)).

Finally, to test whether peak genetic variants may be acting on mitochondrial encoded gene expression via distal associations with genes in the nuclear genome, we performed association analyses between each peak genetic variant and all other nuclear genes not in cis (genes > 1 MB away or on different chromosomes). After correcting for multiple tests, we observe no significant associations (p>0.05 in all cases, Bonferroni correction). Collectively these results suggest that the common mechanisms by which nuclear genetic variation influences mitochondrial encoded gene expression could be either through functional mutations within nuclear genes themselves, or via their effects on the expression of nearby nuclear genes. There is also some evidence that the protein products of some of these genes then enter the mitochondria and bind directly to mitochondrial RNA. Genes identified via these approaches therefore represent the most promising candidates for causal nuclear genes that influence fundamental biological processes taking place in human mitochondria.

**Table 2.** Significant associations between nuclear genetic variants and the expression levels of genes encoded in the mitochondrial genome that replicate in independent tissue matched datasets.
Point-wise permutation P values were generated by extrapolating from the underlying beta distribution (see Materials and methods).

| Tissue | Peak nuclear SNP | Chr | Position | A1 | MAF | MT gene | P value | P value (FDR corrected) | P value (Point-wise Permutation) | Beta | Replication P value | Replication beta | Nuclear gene annotation | Nuclear SNP annotation |
|---|---|---|---|---|---|---|---|---|---|---|---|---|---|---|
| LCLs | rs10205130 | 2 | 44151572 | C | 0.426 | MTND1 | 1.23E-10 | 6.40E-05 | 1.18E-10 | 0.023 | 5.70E-05 | 0.015 | LRPPRC | Intronic |
| Whole Blood | rs10172506 | 2 | 55865469 | C | 0.062 | MTND5 | 4.40E-40 | 2.00E-33 | 1.39E-39 | 0.058 | 2.00E-08 | 0.037 | PNPT1 | Intronic |
| Whole Blood | rs7558127 | 2 | 55866605 | G | 0.064 | MTND6 | 5.26E-32 | 8.30E-26 | 5.29E-32 | 0.070 | 8.70E-07 | 0.043 | PNPT1 | Intronic |
| LCLs | rs2627775 | 2 | 55877113 | T | 0.458 | MTCO1 | 1.33E-09 | 4.10E-04 | 9.95E-10 | 0.017 | 6.30E-04 | 0.013 | PNPT1 | Intronic |
| LCLs | rs62165226 | 2 | 55878339 | C | 0.484 | MTCO3 | 2.79E-08 | 7.60E-03 | 2.71E-08 | −0.018 | 6.80E-06 | −0.018 | PNPT1 | Intronic |
| Whole Blood | rs6973982 | 7 | 45143892 | G | 0.148 | MTCO2 | 7.53E-31 | 8.60E-25 | 1.12E-30 | −0.028 | 2.30E-08 | −0.023 | TBRG4 | Intronic |
| Whole Blood | rs6973982 | 7 | 45143892 | G | 0.148 | MTCO3 | 1.23E-25 | 6.40E-20 | 1.05E-25 | −0.028 | 1.40E-04 | −0.019 | TBRG4 | Intronic |
| Skin (Not sun exposed) | rs2304693 | 7 | 45148667 | A | 0.201 | MTCO3 | 2.84E-12 | 2.10E-05 | 2.93E-12 | −0.027 | 1.80E-03 | −0.026 | TBRG4 | Missense |
| Whole Blood | rs2304694 | 7 | 45148773 | A | 0.148 | MTND1 | 2.34E-17 | 6.10E-12 | 3.19E-17 | 0.015 | 1.60E-08 | 0.024 | TBRG4 | Missense |
| LCLs | rs2304694 | 7 | 45148773 | A | 0.197 | MTND4 | 2.35E-18 | 3.90E-11 | 2.22E-18 | −0.029 | 1.40E-04 | −0.019 | TBRG4 | Missense |
| Subcutaneous Adipose | rs12579998 | 12 | 27861446 | G | 0.174 | MTND1 | 2.48E-23 | 6.50E-17 | 1.09E-23 | −0.032 | 3.20E-05 | −0.032 | - | Intergenic |
| Subcutaneous Adipose | rs12579998 | 12 | 27861446 | G | 0.174 | MTND4 | 2.20E-12 | 1.70E-06 | 1.56E-12 | −0.022 | 4.00E-04 | −0.027 | - | Intergenic |
| Skin (Not sun exposed) | rs11049103 | 12 | 27862081 | A | 0.179 | MTND1 | 7.72E-12 | 2.10E-05 | 6.68E-12 | −0.022 | 8.10E-04 | −0.026 | - | Intergenic |
| Whole Blood | rs74863981 | 20 | 3109875 | A | 0.099 | MTCO1 | 4.24E-11 | 5.40E-06 | 5.25E-11 | −0.024 | 6.60E-05 | −0.022 | UBOX5 | Intronic |

DOI: https://doi.org/10.7554/eLife.41927.008

## Replication and validation of associations

In order to test the robustness of associations between common nuclear genetic variants and mitochondrial gene expression, we tested whether *trans*-genome eQTLs detected in multi-dataset tissues were significant in independent tissue-matched samples (see Materials and methods). In total, 61 eQTLs were found in multi-dataset tissues; to consider the signal replicated we required the association to be between the same variant (or nearest variant in LD ($r^2$ >0.8) if the same variant was not present, or the nearest variant with $r^2$ >0.5 otherwise) and mitochondrial gene in the same tissue type, with the same direction of effect and passing a significance threshold corrected for the number of tests (0.05/61 = 0.00082 in this case). In total we replicate 13/61 (~21.3%) of the mitochondrial *trans*-genome eQTLs (*Figure 3A*, example association shown in *Figure 2B and C*, *Table 2*), and for ten of these we find a link to a potential casual gene through mediation by a nearby nuclear gene or via functional mutations as outlined above (*Table 1*). We also find that an additional 12 associations replicate at the 5% level, and in total 43/61 of the associations show the same direction of effect in replication datasets; larger sample sizes may increase replication rates in these cases.

In order to uncover potential reasons for a lack of replication for some associations, we performed power analysis using the variance explained by each genetic variant on the associated mitochondrial encoded gene expression level in the discovery dataset, together with the replication sample size, and find that ~40.5 associations would be expected to replicate (at p=0.00082). Beyond this, we find significant differences between discovery and replication datasets for the proportion of mapped reads aligning to the mitochondrial genome in whole blood and subcutaneous adipose (Wilcoxon tests, p<0.05 after correcting for multiple tests). It is unclear whether this would influence our ability to replicate associations in these cases, although we note that PEER factors (which we include as covariates in our association analyses) have been shown to correlate with known technical and biological features of RNA sequencing data (*Stegle et al., 2010*; *GTEx Consortium et al., 2017*; *Glastonbury et al., 2018*) and as such should control for some systematic variation across individuals. Even so, given the unexplained lack of replication in some cases, it is possible that false positives may contribute to our results.

To validate our results for one association (rs2304694-*MTND4* in LCLs) using an alternative RNA quantification method, we obtained LCLs with homozygous reference and non-reference genotypes at rs2304694, matched for sex and ethnicity between the two groups, and measured expression levels of *MTND4* using quantitative PCR. We find significant differences in the expression levels of *MTND4* between samples that are homozygous for the reference allele at rs2304694 versus samples that are homozygous for the non-reference allele at rs2304694 (p=0.0325, one-way ANOVA, *Figure 3B*), thus validating the original association with the same direction of effect.

## Links to complex disease

Finally, since genetic variation modulating gene expression may underlie a large proportion of genetic associations with disease (*Nicolae et al., 2010*), we intersected peak mitochondrial *trans*-genome eQTL SNPs, as well as those in strong linkage disequilibrium (LD, $r^2$ >0.8, calculated within our data), with significant associations documented in the NHGRI genome wide association study (GWAS) catalogue and find overlapping variants for two diseases/disease risk traits. First, the peak nuclear genetic variant associated with the expression of *MTCYB* in whole blood (rs782633) is in strong LD with rs782590, a variant that has been linked to systolic blood pressure (a known risk factor for heart disease and stroke) in a study of individuals with metabolic syndrome and controls (*Kristiansson et al., 2012*). We also note that the same peak nuclear genetic variant associated with the expression of *MTCYB* is also in LD with rs1975487 ($r^2$ = 0.84 for Europeans in 1000 Genomes data), a variant that is associated with diastolic blood pressure in a larger GWAS for blood pressure (*Ehret et al., 2016*) (p=2×10$^{-9}$). Rs1975487 was not present in our original analysis due to a missingness rate that was above our threshold for filtering (3%, 2% and 1.7% missing genotype rate in CARTaGENE, TwinsUK and GTEx data respectively). Mitochondrial processes have previously been associated with blood pressure (*Dikalov and Dikalova, 2016*), and given the association here, this may at least partially be modulated though changes in mitochondrial encoded gene expression. The genetic variant associated with mitochondrial encoded gene expression falls within the intron of *PNPT1*, suggesting that this may be the gene of action influencing blood pressure, although further fine mapping and functional work would be required to establish a causal link.

Second, two peak genetic variants associated with the expression of *MTND5* and *MTND6* in whole blood (rs10172506 and rs7558127 respectively) are in strong LD with rs10200159, which has been associated with Vitiligo (*Jin et al., 2016*), a disease that is driven by the functional loss of melanocytes in the skin which leads to a loss of pigmentation. High reactive oxygen species generation and a deficit of the antioxidant network are key processes in Vitiligo, and thus altered mitochondrial function is thought to play a role (*Dell'Anna et al., 2017*). Although we detect significant associations in whole blood, there is suggestive evidence of the same relationships in sun-exposed skin data, with both associations occurring with p<0.05 (p=0.016 and p=0.0027, GTEx data) and the same direction of effect. For both peak nuclear genetic variants the gene of action appears to be *PNPT1*, where we find evidence of significant mediation on the expression of *MTND5* and *MTND6* (*Table 1*).

Genome-wide association studies considering blood pressure were conducted in individuals of Finnish (*Kristiansson et al., 2012*) and European descent (*Ehret et al., 2016*), and the study of Vitiligo was also conducted using individuals of European descent (*Jin et al., 2016*). Since our eQTL analysis included individuals from diverse ancestries (although largely of European descent), we attempted to match LD structure more closely to populations used in the above GWAS associations by re-running mitochondrial encoded eQTL analyses using only samples from individuals of European descent (see Materials and methods). Using the same approach as before, in whole blood data we find that rs782633 remains significantly associated with the expression of *MTCYB* in Europeans (p=$6.33 \times 10^{-11}$ in Europeans, p=$8.58 \times 10^{-11}$ in all samples), rs10172506 is significantly as associated with the expression of *MTND5* in Europeans (p=$4.01 \times 10^{-25}$ in Europeans, p=$5.26 \times 10^{-32}$ in all samples) and rs7558127 is significantly as associated with the expression of *MTND6* in Europeans (p=$1.94 \times 10^{-31}$ in Europeans, p=$4.40 \times 10^{-40}$ in all samples). Furthermore, we find that overlapping mitochondrial-encoded eQTL and GWAS variants are in strong LD in combined European populations surveyed by the 1000 Genomes project ($r^2$ >0.8 in all cases). These results imply that genetic variants associated with mitochondrial encoded gene expression are genuinely in LD with GWAS signals, however some caution should still be applied if populations within Europe are likely to generate further substructure in the data, which we have limited power to disentangle here.

## Discussion

Despite key roles for mitochondria in a range of fundamental biological processes, as well as a wide array of human diseases, knowledge of how the mitochondrial transcriptome is processed across different individuals and tissues on a population scale is incomplete. Using RNA sequencing data for a large number of individuals and across a wide range of tissues, we find considerable variation in mitochondrial gene expression along the mitochondrial genome, across tissues and between individuals. Variation in mitochondrial encoded gene expression profiles is likely important for the cells ability to respond to changing energy demands in specific cell types and environments, and may also play a role in tissue specific disease processes across individuals.

Through integrated analysis of genetic and RNA data, we identify a large number of common nuclear genetic variants associated with mitochondrial encoded gene expression and replicate a substantial fraction of these (~21% after correcting for multiple testing,~41% at nominal 5% with the same direction of effect) in independent tissue-matched datasets. Through mediation analysis and functional genetic variants we identify the potential causal nuclear gene influencing mitochondrial encoded gene expression in 36 cases. A large number of these genes are already known to play a role in mitochondrial processes, and thus validate our findings in a biological context, but also implicate functional mechanisms by which common nuclear genetic variation can act between chromosomes (and indeed, genomes) to influence gene expression. Such *trans*-eQTLs have been notoriously difficult to replicate in humans, and thus the 13 replicated associations identified in this study provide candidates to test the mechanisms associated with genetic variation that acts over large genetic distances.

For some of the potential causal nuclear genes that we identify as being linked to variation in the expression mitochondrial-encoded genes, it is not difficult to speculate on potential mechanisms through which they might act. For example, *MTPAP* (within which we identify a missense mutation associated with the expression of *MTND3* in LCLs) synthesizes the poly(A) tail of mitochondrial transcripts. Since polyadenylation of mitochondrial transcripts is required in many cases to complete the

termination codon and is thought to influence RNA stability (*Rackham et al., 2012*), a functional mutation in this enzyme may lead to variable accumulation of unprocessed mitochondrial transcripts and ultimately influence mitochondrial encoded gene expression levels. Similarly, *TBGR4* (within which we identify a missense mutation associated with the expression of multiple mitochondrial genes in multiple tissues) is known to process mitochondrial precursor transcripts and stabilize some mature mitochondrial messenger RNAs (*Boehm et al., 2017*), thus having obvious links to changes in mitochondrial gene expression. These findings lay the foundation for future work to functionally validate the causal role of these genetic variants.

Beyond this, we also identify nuclear genes through mediation analysis that have not previously been linked with mitochondrial gene expression. These results potentially point to novel roles for these proteins and thus may be important new targets in the context of mitochondrial disease in cases where it has thus far been difficult to identify causal mutations in patients. Examples that may be interesting for further study include *ZFP90*, a zinc finger protein that modulates nuclear gene expression. *ZFP90* transgenic mice show altered expression of genes involved in oxidative phosphor-ylation and fatty acid elongation in mitochondria compared to wild type littermates (*Yang et al., 2009*), pointing to a potential role in mitochondrial processes. Similarly, *CCM2* is involved in the stress-activated p38 mitogen-activated protein kinase (MAPK) signalling cascade and is thought to localize to the mitochondria. CCM proteins are implicated in Cerebral Cavernous Malformation and accumulating evidence points to a role for these proteins in processes related to mitochondrial func-tion, including cellular responses to oxidative stress and autophagy (*Retta and Glading, 2016*).

Finally, the common genetic variants we identify here as associated with mitochondrial encoded gene expression profiles across individuals potentially have downstream functional consequences that influence disease processes and risk. We find some evidence for this, as nuclear genetic varia-tion associated with variable mitochondrial encoded gene expression is linked to mutations that have been implicated in blood pressure and Vitiligo, yet further study of these genes is required to identify the causal mechanisms that influence how mitochondrial RNA is processed in the cell and how dysregulation of these mechanisms may cause disease. Combined, these data now serve as a frame of reference for mitochondrial disease researchers who wish to consider how patient samples may vary in mitochondrial gene expression versus a healthy cohort in the relevant tissue type, and for the community as whole interested in the genes and genetics of fundamental processes taking place in mitochondria and the genetic architecture of gene expression.

## Materials and methods

### Data

Raw human RNA sequencing and genotyping data were obtained through application to five inde-pendent sequencing projects:

*CARTaGENE*: CARTaGENE is a healthy cohort of individuals aged between 40 and 69 from Que-bec, Canada. Whole blood, 100 bp paired-end RNA sequencing and genotyping data (Illumina Omni 2.5M arrays) for 911 individuals were obtained from the CARTaGENE project (*Awadalla et al., 2013*; *Hodgkinson et al., 2014*) through application to the data access committee (instructions are available at www.cartagene.qc.ca). Samples with multiple sequencing runs were merged prior to alignment.

*TwinsUK*: 50 bp paired-end RNA sequencing data from 391 whole blood samples, 685 subcutane-ous adipose samples, 672 non-sun exposed skin samples and 765 LCL samples (*Buil et al., 2015*), as well as accompanying genotyping information (obtained from either Illumina HumanHap300 and HumanHap610Q arrays), were derived from a mix of unrelated samples and monozygotic and dizy-gotic twin pairs through application to the TwinsUK data access committee and then downloaded from the European Genome-Phenome archive (https://ega-archive.org) through study ID EGA S00001000805.

*GTEx (Genotype-Tissue Expression) Project:* 75 bp paired-end RNA sequencing data from 44 tis-sue/cell types from up to 572 individuals (*GTEx Consortium et al., 2017*), along with accompanying genotyping data (obtained from either Illumina Omni5M and Omni2.5M arrays) were obtained by application to dbGaP through accession number phs000424.v6.p1. Tissues were selected if the organ they were obtained from had at least 100 samples. In cases where samples had multiple

sequencing experiments for a given individual and tissue, we selected the dataset containing the highest number of raw sequencing reads.

*NIMH (National Institute of Mental Health) Genomics Resource*: 50 bp single end RNA sequencing data and matched genotyping data (Illumina HumanOmni1-Quad BeadChip) from 937 whole blood samples (*Battle et al., 2014*; *Mostafavi et al., 2014*) from the Depression Genes and Networks study were obtained via transfer from external hard drives after application to the data access committee (through www.nimhgenetics.org).

*Geuvadis Project*: 75 bp paired end RNA sequencing data from 462 LCL samples (*Lappalainen et al., 2013*) were downloaded from the European Nucleotide Archive under submission number ERA169774. Accompanying genetic variants from whole genome sequencing data (which were generated as part of the 1000 Genomes Project (*Abecasis et al., 2012*)) were downloaded from the 1000 genomes FTP site. We used phase three data that was phased and imputed (v5a.20130502).

## Processing of RNA sequencing data

All RNA sequencing data derived from different projects were processed in the same way to ensure comparability across analyses. Raw RNA sequencing reads (fastq format) from 13,261 individual samples were trimmed for adaptor sequences, terminal bases with nucleotide quality below 20 and poly (A) tails > 4 bp in length, before being aligned to a reference genome (1000G GRCh37 reference, which contains the mitochondrial rCRS NC_012920.1) with STAR 2.51a (*Dobin et al., 2013*), using two-pass mapping, version 19 of the Gencode gene annotation and allowing for 1/18*read_length mismatches, rounded down to the nearest integer. Following this, in order to minimize the likelihood of incorrectly placed reads (particularly those associated with NUMT sequences), we used a stringent filtering pipeline, focusing only on reads that were properly paired and uniquely mapped. After mapping we removed low quality samples that had either <10 thousand reads mapping to the mitochondrial genome,<5 million total mapped reads,>30% of reads mapping to intergenic regions,>1% total mismatches or >30% reads mapping to ribosomal RNA using in house scripts and RNAseQC (*DeLuca et al., 2012*). To calculate transcript abundances, we used HTseq (*Anders et al., 2015*) with the 'intersect non-empty' model and version 19 of the Gencode gene annotation, before converting raw counts to transcripts per million (TPM). We plotted the $\log_{10}$ transformed distributions of all genes with mean TPM >2 per sample and removed visual outlier samples. We also calculated principle components using the same data and removed outlier samples. Finally, samples were only included in analyses if they had accompanying high quality genotyping information (see below) and there were at least 70 samples available for analysis within each tissue/dataset; in total after matching samples to genotyping data and quality control filtering we were left with 11,371 RNA sequencing datasets for analysis. We focused on mitochondrial encoded protein coding and ribosomal RNA genes only, since transfer RNAs showed lower sequencing coverage overall and were not expressed highly in all tissues and datasets. For analysis of mitochondrial encoded gene expression variation across genes and datasets, for TwinsUK data we used only unrelated samples (which involved picking one of each twin pair at random and combining these with unrelated samples). For NIMH samples, which were derived from 454 depression cases and 454 controls, we tested whether disease status may affect our results by comparing TPM values for mitochondrial-encoded genes between the two groups; in all cases we find no significant differences (Wilcoxon test, p>0.05 in all cases after correction for multiple testing).

## Processing of genotyping data

Genotyping data from different arrays and sequencing studies were processed separately. For TwinsUK data, only one twin from each twin pair was genotyped and thus processed, with data duplicated to represent the missing twin pair after quality control and filtering. Genotyping quality control and calculation of genetic principle components for Twins data was thus performed only on unrelated samples. Within each dataset, samples with high relatedness (>0.125), high SNP heterozygosity (visual outliers), non-matching sex, ambiguous X-chromosome homozygosity estimates or high SNP missingness (>5%) were removed. Autosomal SNPs were flipped to the positive strand and those with minor allele frequency (MAF) >1%, in Hardy Weinberg equilibrium (p>0.001) and not missing in more than 1% of individuals were then phased with shapeit2 (*Delaneau et al., 2013*) using

no reference panel and default settings. Problematic sites were removed and remaining SNPs were used for imputation in 2 MB intervals using impute2 (*Howie et al., 2009*) with default settings, incorporating the 1000 Genomes phase three reference panel. Imputed data were then hard-called to produce genotypes at each site with a threshold of 0.9 and SNPs with information score lower than 0.8 were removed. Data from different arrays within each study were then merged and filtered to keep bi-allelic variants with minor allele frequency (MAF) >5%, in Hardy Weinberg equilibrium (p>0.001) and not missing in more than 1% of individuals for downstream analysis. After processing, we calculated genetic principal components and removed outlier samples by visual inspection. For Geuvadis data we used whole genome sequencing variant calls from the 1000 Genomes project (*Abecasis et al., 2012*). As such, these samples did not undergo phasing and imputation within our pipeline, but were filtered in the same way as genotyping data after this stage of the analysis.

## Association analyses

Expression QTL mapping was performed within each tissue and sequencing dataset. In each case, TPM values for thirteen mitochondrial encoded protein coding genes and two mitochondrial encoded ribosomal RNA genes were extracted before being $log_{10}$ transformed (*Supplementary file 4*). Mitochondrial encoded gene expression distributions were median normalized, before outlier values were removed per gene (defined as three interquartile ranges above or below the upper and lower quartile respectively). To control for unidentified confounding factors in RNA sequencing data, we calculated PEER factors (*Stegle et al., 2010*) per dataset using all genes (nuclear and mitochondrial) that had a mean TPM >2. For genotyping data, we restricted the data to only those samples that had corresponding mitochondrial encoded gene expression values for the given dataset and calculated genetic principle components on this reduced set in each case. We then performed association analyses on each tissue and dataset using a linear model within PLINK (*Purcell et al., 2007*) for unrelated samples. For twin data, we calculated the relatedness matrix of samples before conducting association analyses with GEMMA (*Zhou and Stephens, 2012*). In each case we included sex, five genetic principle components, 5 or 10 PEER factors (five for samples sizes < 100, ten for sample sizes >= 100) and sequencing/genotyping batch (where applicable) as covariates. For TwinsUK data, the genotyping array was included as the batch covariate and sex was omitted as all samples were derived from females. For CARTaGENE data, which was original sequenced at higher and lower coverage as part of discovery and replication phase data respectively (*Hodgkinson et al., 2014*), the sequencing phase was included as the batch covariate. For GTEx data, where two different genotyping arrays were used, the genotyping array covariate correlated highly with one of the first genetic principle components for all tissues (|r| > 0.8 in all cases) and was therefore not included in the linear model. After analysis, QQ plots were visually assessed and show no skew. QQ plots for discovery associations that replicate at the nominal 5% level are shown in *Figure 2—figure supplement 1*. False discovery correction (Benjamini-Hochberg) was applied to raw p-values within each dataset by merging all genes (15) and genetic variants in each case, following the approach applied by the GTEx consortium (*GTEx Consortium et al., 2017*).

To calculate P-values via permutation analysis, for each association that we originally identified as being significant at FDR 5% (64 variant-gene pairs), we performed 100,000 point-wise permutations for the relevant tissue type, mitochondria-encoded gene and nuclear genetic variant by randomly shuffling phenotypes. In each case, we then collected the test statistic across all 100,000 permutations to generate a null distribution, and compared our observed test statistic against this to calculate an empirical P-value. For tissue types with multiple datasets (Whole Blood and LCLs) we performed permutations per dataset, combined these within a meta-analysis, and then derived the null distribution from the meta-analysis results. In each case, we also then followed the approach outlined in *Ongen et al. (2016)* to calculate a more precise P-value by estimating the underlying beta distribution of the null distribution via maximum likelihood (using the 'ebeta' function within the R package 'EnvStats').

Additionally, we also calculated the family-wise error rate on the gene level for each association originally detected at FDR 5%. To do this, we performed 200 random permutations across all nuclear genetic variants for the relevant mitochondria-encoded gene and tissue type, and then calculated the null distribution by selecting the largest test statistic per permutation across all nuclear genetic variants. To calculate the overall family wise error rate, we repeated this again, this time selecting the largest test statistic across all nuclear genetic variants and all 15 mitochondria-encoded genes

per permutation to generate the null distribution in the relevant tissue type. For the calculation of both family-wise error rates, we repeated the approach outlined in *Ongen et al. (2016)* to obtain a more precise P-value by extrapolating from the beta-distribution generated from the null. P values generated across all methods are shown in **supplementary file 1**.

NUMT sequences were obtained from the UCSC genome browser track named 'numtS', and were generated by *Simone et al. (2011)*, who used blastN to map nuclear chromosomes to the mitochondrial genome, setting the e-value threshold to 0.001. Sequences in this database range from 31 to 14904 bp in length, with a similarity percentage ranging between 63% and 100%, thus the approach has the potential to tolerate a large number of mismatches between nuclear and mitochondrial sequences. To test whether any 50 bp segments of mitochondrial genes also aligned to nuclear genes, we followed the approach defined in *Saha and Battle (2018)*. Specifically, we took all 50 bp k-mers from each mitochondrial encoded gene and then aligned these sequences to the nuclear genome using bowtie v1.22 (*Langmead et al., 2009*), allowing for up to two mismatches and reporting all alignments. For each nuclear genetic variant associated with a mitochondrial encoded gene, we then tested whether any of the 50 bp k-mers from the mitochondrial encoded gene aligned within a nuclear gene whose transcription start site fell within 1 MB of the corresponding nuclear genetic variant.

For tissue types with multiple independent datasets, we defined discovery and replication datasets. Discovery datasets were chosen as the dataset with the largest starting sample size for each given tissue, with the replication dataset as the second largest. For whole blood, where four independent datasets were available, we performed meta analysis within PLINK using a fixed affects model, combining data from the CARTaGENE project, GTEx and TwinsUK for the discovery phase, and then used NIMH data for replication. For LCLs, where three independent datasets were available, we performed meta analysis combining data from the Twins and GTEx for the discovery phase, and then used Geuvadis data for replication. For Subcutaneous adipose and non-sun exposed skin, we used TwinsUK data for discovery and GTEx data for replication. For all other tissues, only a single dataset was available, and so no replication analysis was performed. In all association analyses we defined the peak SNP as the genetic variant with the lowest p-value within a block of 1 MB, and tested for replication using the exact same SNP where available (using the nearest SNP in LD ($r^2$ >0.8) if the exact match was not present, followed by the nearest SNP with $r^2$ >0.5 otherwise). We used the same approach when comparing association signals across tissues. To perform power calculations, we obtained the correlation coefficient ($r^2$) between the genetic variant and the expression of the associated mitochondrial encoded gene in the relevant discovery dataset (or largest dataset where the genetic variant is present, if multiple datasets are available for the tissue). We then used a power calculator (*Purcell et al., 2003*), specifying our estimate for the variance explained by the genetic variant ($r^2$), the minor allele frequency, replication sample size and the significance threshold (0.05/61) in each case. Following this, we summed power values across all 61 associations.

We also repeated all association analyses after using mitochondrial library size (all reads mapping to the mitochondrial) to calculate TPM for mitochondrial genes, rather than total library size. We tested this approach as a way to remove the effects of variable mitochondrial copy number and poly-cistronic transcription rate, however in all cases we obtained very similar results to those obtained using the method outlined above. Additionally, we also repeated all analyses shown in *Figure 1* using mitochondrial reads to normalize gene expression values; again we find very similar results.

It has recently been shown that the post-mortem interval (PMI) appears to influence gene expression patterns in GTEx data (*Ferreira et al., 2018*). As such, to test for an effect in our data, we repeated association analyses for significant associations discovered in GTEx data and including PMI as a covariate (where PMI data were available). In both cases, we find that the P-values do not change dramatically (Atrial appendage (heart), rs11811165-*MTND4L*, original raw P value: $5.09 \times 10^{-10}$, P value including PMI as a covariate: $3.50 \times 10^{-9}$; Tibial nerve, rs932345-*MTND4L*, original raw P value: $6.47 \times 10^{-10}$, P value including PMI as a covariate: $7.57 \times 10^{-10}$).

## Functional annotation and links to complex disease

In order to identify the potential causal nuclear gene associated with mitochondrial encoded gene expression, we identified genes associated with the peak eQTL variant in the following ways. First, if the peak variant was a missense mutation, we assumed that its mode of action was via functional

changes in the gene it was located in. Second, for non-coding mutations, we tested whether non-coding peak variants fell in enhancer regions using chromatin state predictions obtained from 128 cell types within the Roadmap Epigenetic project (*Kundaje et al., 2015*), using matched tissue data as outlined in the GTEx project (*GTEx Consortium et al., 2017*), and compared this against a set of random genetic variants matched for minor allele frequency, distance from transcription start site and genome annotation (using 1000 random sets to generate a P-value). Third, for non-coding peak variants, we tested for mediation via the expression of nuclear genes located near to the peak SNP. To do this, for each tissue we used the largest dataset available and restricted our analysis to unrelated samples (for TwinsUK data, this involved picking one of each twin pair at random and combining these with unrelated samples). Within each dataset we then again tested for a significant correlation between the peak SNP and the expression of the mitochondrial gene in question ($p<0.05$, linear model, t-test of regression coefficient), as well as a significant correlation between the peak SNP and the expression of any nuclear gene within 1 MB of the variant ($p<0.05$, linear model, t-test of regression coefficient). For genes/variants passing these criteria, we then tested whether the expression of the nuclear gene significantly mediated the relationship between the peak nuclear variant and the mitochondrial encoded gene expression using the module 'mediation' (testing significant mediation of the initial association via bootstrapping, requiring an average causal mediation effect with $p<0.05$ after FDR correction) within R. To prioritize potential causal genes within this framework, we first selected nuclear genes with a known role in mitochondrial processes (any gene listed in the Mitocarta database (*Calvo et al., 2016*), shown to influence mitochondrial RNA processing (*Wolf and Mootha, 2014*) or listed as being involved in mitochondrial disorders in the Genomics England PanelApp - https://panelapp.genomicsengland.co.uk), before moving on to any other nuclear gene. Finally, we tested whether non-coding peak variants were associated with the expression of more distal genes (those whose transcription start site was >1 MB away, or on another chromosome) within a linear model (and meta-analysis where relevant) including the same datasets, methods and covariates as the original discovery analysis.

In order to identify whether genetic variants associated with mitochondrial encoded gene expression may play a role in complex disease, we first identified any SNP in linkage disequilibrium ($r^2 >0.8$, calculated using datasets and samples used in this study) with peak eQTL SNPs in any of the datasets used for the tissue type in which the association was identified. We then tested whether any of these variants overlapped with significant associations documented in the NHGRI GWAS catalogue (for association where $p<5e-8$). To test whether associations between nuclear genetic variants and mitochondrial encoded gene expression that overlap GWAS signals are significant in individuals of European descent, we plotted the first two genetic principal components against those derived from 1000 genomes samples with known ancestry for any dataset that had associated RNA sequencing data from whole blood. We then selected samples that clustered with Europeans in 1000 genomes data by visual inspection and re-ran association analyses as before for whole blood data from CARTaGENE, TwinsUK and GTEx, before performing meta-analysis to calculate P-values.

## Validation

In order to validate the association between rs2304694 and expression levels of *MTND4* in LCLs, we obtained ten LCL samples carrying the homozygous reference genotype and ten LCL samples carrying the homozygous non-reference genotype for rs2304694 from the Coriell Institute for Medical Research, matched between the two genotype groups for sex and ethnicity (*Supplementary file 5*). The following cell lines were obtained from the NIGMS Human Genetic Cell Repository at the Coriell Institute for Medical Research: GM11919, GM11932, GM12003, GM12414, GM12717, GM12842. The following cell lines were obtained from the NHGRI Sample Repository for Human Genetic Research at the Coriell Institute for Medical Research: GM20582, GM20822, HG00118, HG00254, HG00284, HG00290, HG01524, HG01625, HG01631, HG01777, HG01800, HG01804, HG01812, HG01815. Cultures were tested as standard by Coriell Cell Repositories before shipping and found free of mycoplasma, and microsatellite profiling was used to confirm identity (see 'Quality Control' at www.coriell.org). Cells were handled as per supplier's instructions. Total RNA was extracted using the RNeasy kit (Qiagen) according to the manufacturer's instructions. 1 ug total RNA was pretreated with 2 units of Turbo DNase (Fisher Scientific) and subsequently reverse-transcribed using the ProtoScript First Strand cDNA synthesis kit (New England BioLabs) with random primers. The first strand reaction was diluted five fold with deionised water and 1% (vol/vol) was used as template

for each real-time PCR (RT-PCR) reaction. RT-PCR was carried out using QuantiNova SYBR Green (Qiagen) and a StepOnePlus RT-PCR System (Applied Biosystems). Primers used were as follows: *GAPDH* (F: TCTGCTCCTCCTGTTCGACA, R: AAAAGCAGCCCTGGTGACC), *MTND4* (F: CAC TAAACATTCTACTACTCACTCTC, R: GGAGTCATAAGTGGAGTCCGTA). Expression levels of *MTND4* were determined after normalization to *GAPDH* (theoretical quantities), and two technical qPCR replicates were performed per sample before being averaged. Outlier values were removed (defined as three interquartile ranges above or below the upper and lower quartile respectively) within each genotypic category, leaving 19 samples for analysis. This association was chosen for replication analysis since it is associated with mitochondrial encoded gene expression across multiple tissue types and is significantly associated with *MTND4* in a dataset and tissue type for which we had access to the relevant biological material (Geuvadis dataset, LCLs).

## Acknowledgements

Anonymized processed mitochondrial encoded gene expression matrices are available in *supplementary file 4* and from the Gene Expression Omnibus under accession GSE125013. We thank the CARTaGENE platform, GTEx, TwinsUK, the NIMH Genomics Resource and the Geuvadis project for use of RNA sequencing data. We thank Youssef Idaghdour for comments on the manuscript. This work was supported by the Biotechnology and Biological Sciences Research Council (BBSRC) through award BB/R006075/1. AH holds a Medical Research Council (MRC) eMedLab Medical Bioinformatics Career Development Fellowship, funded from award MR/L016311/1. AH also holds a WHRI-Academy Marie Curie (COFUND) Fellowship and the research leading to these results has received funding from the People Programme (Marie Curie Actions) of the European Union's Seventh Framework Programme (FP7/2007-2013) under REA grant agreement n° 608765. Work presented here reflects only the author's views and not the views of the European Commission. ATA is supported by the Generation Trust. GC is supported by a BBSRC Project Grant (BB/R006075/1). KSS is supported by an MRC Project Grant (MR/L01999X/1). VCS is supported by a career development award from the MRC (MR/M009343/1). LB is supported by the Guy's and St. Thomas' Charity (MAJ110901) and is a King's College London member of the MRC Doctoral Training Partnership in Biomedical Sciences. The research was supported by the National Institute for Health Research (NIHR) Biomedical Research Centre based at Guy's and St Thomas' NHS Foundation Trust and King's College London. The views expressed are those of the authors and not necessarily those of the NHS, the NIHR or the Department of Health. For GTEx data: the Genotype-Tissue Expression (GTEx) Project was supported by the Common Fund of the Office of the Director of the National Institutes of Health (commonfund.nih.gov/GTEx). Additional funds were provided by the NCI, NHGRI, NHLBI, NIDA, NIMH, and NINDS. Donors were enrolled at Biospecimen Source Sites funded by NCI\Leidos Biomedical Research, Inc subcontracts to the National Disease Research Interchange (10XS170), Roswell Park Cancer Institute (10XS171), and Science Care, Inc (X10S172). The Laboratory, Data Analysis, and Coordinating Center (LDACC) was funded through a contract (HHSN268201000029C) to the The Broad Institute, Inc Biorepository operations were funded through a Leidos Biomedical Research, Inc subcontract to Van Andel Research Institute (10ST1035). Additional data repository and project management were provided by Leidos Biomedical Research, Inc (HHSN261200800001E). The Brain Bank was supported supplements to University of Miami grant DA006227. Statistical Methods development grants were made to the University of Geneva (MH090941 and MH101814), the University of Chicago (MH090951,MH090937, MH101825, and MH101820), the University of North Carolina - Chapel Hill (MH090936), North Carolina State University (MH101819),Harvard University (MH090948), Stanford University (MH101782), Washington University (MH101810), and to the University of Pennsylvania (MH101822). For NIMH data: data were provided by Dr. Douglas F Levinson (dflev@stanford.edu). We gratefully acknowledge the resources were supported by National Institutes of Health/National Institute of Mental Health Grants 5RC2MH089916 (PI: Douglas F Levinson, M.D.; Co-investigators: Myrna M Weissman, Ph.D., James B Potash, M.D., MPH, Daphne Koller, Ph.D., and Alexander E Urban, Ph.D.) and 3R01MH090941 (Co-investigator: Daphne Koller, Ph.D.). For TwinsUK data: The TwinsUK study was funded by the Wellcome Trust and European Community's Seventh Framework Programme (FP7/2007-2013). The TwinsUK study also receives support from the National Institute for Health Research (NIHR)- funded

BioResource, Clinical Research Facility and Biomedical Research Centre based at Guy's and St Thomas' NHS Foundation Trust in partnership with King's College London.

## Additional information

### Funding

| Funder | Grant reference number | Author |
|---|---|---|
| Medical Research Council | MR/L016311/1 | Alan Hodgkinson |
| Biotechnology and Biological Sciences Research Council | BB/R006075/1 | Guillermo Carbajosa Alan Hodgkinson |
| People Programme (Marie Curie Actions) of the European Union's Seventh Framework Programme | FP7/2007-2013 under REA grant agreement number 608765 | Alan Hodgkinson |
| The Generation Trust | | Aminah T Ali |
| Medical Research Council | MR/L01999X/1 | Kerrin S Small |
| Medical Research Council | MR/M009343/1 | Vlad C Seitan |
| Guy's and St Thomas' Charity | MAJ110901 | Lena Boehme |

The funders had no role in study design, data collection and interpretation, or the decision to submit the work for publication.

### Author contributions

Aminah T Ali, Data curation, Formal analysis; Lena Boehme, Validation, Methodology; Guillermo Carbajosa, Formal analysis, Methodology; Vlad C Seitan, Supervision, Methodology, Writing—review and editing; Kerrin S Small, Conceptualization, Methodology, Writing—review and editing; Alan Hodgkinson, Conceptualization, Data curation, Formal analysis, Supervision, Funding acquisition, Investigation, Visualization, Methodology, Writing—original draft, Project administration, Writing—review and editing

### Author ORCIDs

Aminah T Ali (iD) http://orcid.org/0000-0003-1089-9278
Lena Boehme (iD) http://orcid.org/0000-0001-7593-7533
Vlad C Seitan (iD) http://orcid.org/0000-0002-4546-340X
Kerrin S Small (iD) http://orcid.org/0000-0003-4566-0005
Alan Hodgkinson (iD) http://orcid.org/0000-0003-1636-491X

### Decision letter and Author response

Decision letter https://doi.org/10.7554/eLife.41927.025
Author response https://doi.org/10.7554/eLife.41927.026

## Additional files

### Supplementary files

• Supplementary file 1. Significant associations between nuclear genetic variants and the expression levels of genes encoded in the mitochondrial genome. See Materials and methods for a description of the calculation of different permutation P values.
DOI: https://doi.org/10.7554/eLife.41927.009

• Supplementary file 2. Details of trans-genome associations when calculating TPM values for mitochondrial-encoded genes using the total number of reads mapping to the mitochondrial genome.
DOI: https://doi.org/10.7554/eLife.41927.010

• Supplementary file 3. Replication P-values and Beta Coefficients for mitochondrial trans-genome eQTLs in all tissue types.

DOI: https://doi.org/10.7554/eLife.41927.011

• Supplementary file 4. Gene expression matrices for all dataset used in the study. Values show log-transformed (TPM +1) values. After transformation, outlier values were removed (three inter-quartile ranges above/below the upper/lower quartiles respectively).
DOI: https://doi.org/10.7554/eLife.41927.012

• Supplementary file 5. Population, sex and allele information for cell lines used in qPCR validation experiments.
DOI: https://doi.org/10.7554/eLife.41927.013

• Supplementary file 6. Forest plots for each peak variant detected in association analyses. Each plot contains Beta estimates and confidence intervals for each of the datasets and tissues considered in the study.
DOI: https://doi.org/10.7554/eLife.41927.014

• Transparent reporting form
DOI: https://doi.org/10.7554/eLife.41927.015

## Data availability

Anonymized processed mitochondrial encoded gene expression matrices are available in Supplementary file 2 and from the Gene Expression Omnibus under accession GSE125013. Data from CARTaGENE (Awadalla et al., 2013) and NIMH sequencing data from the Depression Genes and Networks study (Battle et al., 2014) used in this work are available through request. Requests for access first need to be approved by a data access committee (further information can be found here https://www.cartagene.qc.ca/en/researchers/access-request and here https://www.nimhgenetics.org/request-access/how-to-request-access).

The following dataset was generated:

| Author(s) | Year | Dataset title | Dataset URL | Database and Identifier |
| --- | --- | --- | --- | --- |
| Ali AT, Boehme L, Carbajosa G | 2019 | Nuclear Genetic Regulation of the Human Mitochondrial Transcriptome | http://www.ncbi.nlm.nih.gov/geo/query/acc.cgi?acc=GSE125013 | NCBI Gene Expression Omnibus, GSE125013 |

The following previously published datasets were used:

| Author(s) | Year | Dataset title | Dataset URL | Database and Identifier |
| --- | --- | --- | --- | --- |
| Buil A, Brown AA, Lappalainen T, Vinuela A, Davies MN | 2015 | Gene-gene and gene-environment interactions detected by transcriptome sequence analysis in twins | https://ega-archive.org/studies/EGAS00001000805 | European Genome-Phenome Archive, EGAS00001000805 |
| GTEx Consortium | 2017 | Common Fund (CF) Genotype-Tissue Expression Project (GTEx) | https://www.ncbi.nlm.nih.gov/projects/gap/cgi-bin/study.cgi?study_id=phs000424.v6.p1 | NCBI dbGaP, phs000424.v6.p1 |
| Lappalainen T | 2013 | Geuvadis Project | https://www.ebi.ac.uk/ena/data/view/ERA169774 | European Nucleotide Archive, ERA169774 |

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
