## [Decision Letter]

Thank you for submitting your article "Nuclear Genetic Regulation of the Human Mitochondrial Transcriptome" for consideration by eLife. Your article has been reviewed by three peer reviewers, and the evaluation has been overseen by a guest Reviewing Editor and Mark McCarthy as the Senior Editor. The following individual involved in review of your submission has agreed to reveal their identity: Dan Arking (Reviewer #1).

The reviewers have discussed the reviews with one another and the Reviewing Editor has drafted this decision to help you prepare a revised submission. As you can see, we have listed a set of "essential revisions" that represent the consensus assessment of the editors and reviewers.

Summary:

The manuscript describes a study of the regulation of the mitochondrial transcriptome, in particular identifying nuclear genetic variants that are associated with expression of mitochondrial transcripts. They identify such associations for fourteen out of fifteen mitochondrial genes and a total of 64 trans-genome associations. Replication between studies is observed, and they discuss the role of genetic variation affecting mitochondrial gene expression in complex disease.

The reviewers appreciated the importance of the findings for understanding mitochondrial gene expression and found the manuscript generally clear. However, to support the findings, some additional analyses are desired along with adding to the biological interpretation of the results and the discussion of some caveats.

Essential revisions:

1) In order to achieve a higher level of biological understanding and impact of the results, it is necessary to more fully characterize the discoveries, particularly those that are not previously implicated in known mitochondrial regulatory processes. This could include an analysis of the variability in expression of those factors; a more systematic analysis of whether those factors are shuttled back to the nucleus or localized with the mitochondria; and analysis of the 45 QTLs that are not missense mutations. For instance, are they in known enhancer regions that may have impacts on multiple mt-regulatory factors? Are the same QTLs (either SNPs or linked SNPs) also associated with nuclear encoded mitochondrial genes that have interconnected signaling pathways with mt encoded genes. Are the gene expression levels of these two sets of mt genes associated with each other across individuals and could there be any regulatory feedback mechanisms across these sets of genes? Not every one of these questions will necessarily yield an interesting result, but some further expansion on biological interpretation is necessary.

2) Replication rate overall is low, which merits exploration. Is this due to power, technical variability, biological variability, or a higher false discovery rate than originally estimated? It is necessary to show analysis attempting to disentangle this through power calculation, simulation, and any relevant demonstration of biological or technical confounders. Could it be driven by differences in mt copy number, or cell state (leading to differences in mt-liked metabolic activity)?

Given low replication rates and other concerns, it is noted that FDR may be susceptible to artifacts, and in general it is necessary to demonstrate that the FDR is well calibrated. The suggested method is to use permutation analysis (permuting a trait, leaving linkage disequilibrium intact) to establish an empirical null. In addition, please include raw p-values along with FDR for detected associations.

3) In order to rule out false positive trans associations due to alignment error, it is necessary to evaluate whether there is sequence similarity between the candidate mt eQTL target genes and the nuclear regions of the genome, particularly for regions near the candidate eQTL variants. If there is sequence similarity, these could represent cis-eQTLs or nuclear eQTLs for genes that simply have some reads mis-mapping to mt. This is a similar problem to probe cross-hybridization observed for microarrays, and has been observed for RNA-seq false positive trans-eQTLs in GTEx and other studies.

4) It is necessary to address questions regarding ancestry and linkage disequilibrium, particularly considering diverse ancestries represented in the RNA-seq data and potential differences with those in the GWAS data used. The manuscript evaluates SNPs in NHGRI GWAS catalogue in strong LD with the SNPs that control mtDNA gene expression. What LD reference is used? Is it the European LD reference calculated from 1000 genomes, or is it the in-sample LD from the datasets with RNAseq data? What were the populations (ancestries) in which the GWAS was performed? In other words, is the LD between SNPs calibrated for the GWAS or for the eQTL analyses? While it may not be possible to match LD perfectly between GWAS and eQTL analyses, these potential issues should be mentioned and some attempt made to determine if they are problematic. If the individual data is available, it is necessary to discuss single/per population analysis to show that their results are real and not artifacts of population structure and also likely consistent across populations. This would involve simply sectioning out the samples from each single population, and rerunning their analysis. The associations are not expected to be significant in the single population analysis since they'll have much smaller sample sizes, but the effect sizes and directions should be correlated. If this is not possible, then caveats of population structure need to be mentioned and the claims toned down accordingly.

---

## [Author Response]

Essential revisions:1) In order to achieve a higher level of biological understanding and impact of the results, it is necessary to more fully characterize the discoveries, particularly those that are not previously implicated in known mitochondrial regulatory processes. This could include an analysis of the variability in expression of those factors; a more systematic analysis of whether those factors are shuttled back to the nucleus or localized with the mitochondria; and analysis of the 45 QTLs that are not missense mutations. For instance, are they in known enhancer regions that may have impacts on multiple mt-regulatory factors? Are the same QTLs (either SNPs or linked SNPs) also associated with nuclear encoded mitochondrial genes that have interconnected signaling pathways with mt encoded genes. Are the gene expression levels of these two sets of mt genes associated with each other across individuals and could there be any regulatory feedback mechanisms across these sets of genes? Not every one of these questions will necessarily yield an interesting result, but some further expansion on biological interpretation is necessary.

We now perform additional analyses to better understand the impact of our results. First, as suggested, we tested whether non-coding variants are enriched in enhancer regions using chromatin state predictions obtained from 128 cell types within the Roadmap Epigenetic project. We find that none of the non-coding nuclear genetic variants associated with mitochondrial encoded gene expression fall in enhancer regions in tissue matched cell types (data available for 44 of the 49 variants), which is not different to that expected by chance (P=0.676, using randomly selected variants matched for minor allele frequency, distance to nearest transcription start site and annotation). Under the assumption that associations between nuclear genetic variants and mitochondrial encoded gene expression occur ubiquitously across the body (and we don’t have the power to detect these associations in all tissues), we tested for the presence of peak variants in enhancer regions in any cell type. In total, 24 variants fall in enhancer regions, which is again not significantly different from random (P=0.691, using randomly selected variants as before).

Second, since we already analyse the impact of non-coding peak variants on nuclear gene expression in *cis*, we also test whether peak variants act over larger genetic distances and are associated with the expression of nuclear genes in *trans*. Performing association analyses between all non-coding peak variants and the expression of all nuclear genes whose transcription start sites are more than 1MB away (or on different chromosomes), we find no significant associations after Bonferroni correction.

Our existing results show that some peak variants fall at functional positions within known nuclear encoded mitochondrial genes. We have also shown that many non-coding peak variants are not only associated with the expression of nearby nuclear genes, but also that the expression of these nuclear genes correlate with the expression of the corresponding mitochondrial encoded gene, and moreover, this correlation explains a significant proportion of the originally detected relationship between the peak nuclear variant and the expression of the mitochondrial encoded gene (mediation analysis). Combining the results from the two new analyses suggested by the reviewers/editors (where we see no evidence of peak variants falling in enhancer regions or influencing the expression of more distal genes), it seems that the mechanisms of action driving the *trans*-genome associations that are most consistent with our results are either directly functional on gene (through a missense mutation), or acting via the expression of a gene in *cis* (identified via mediation analysis) where the peak variant is either not causal itself (but is in LD with the causal mutation) or influences the gene via a mechanism not associated with the actions of known enhancers.

Third, given the above, we further annotate genes identified through functional mutations or mediation analysis via Mitominer (Smith and Robinson, 2016), an integrated resource for mitochondrial localisation and functional evidence. Of the genes identified through functional mutations (*TBRG4, LONP1, MTPAP*), there is evidence that all three proteins are targeted to the mitochondria, and mass spectrometry experiments have identified evidence of these proteins in mitochondria. Furthermore, *TBRG4* and *MTPAP* contain RNA binding domains (Wolf and Mootha, 2014), and so may influence mitochondrial RNA directly. For genes identified through mediation analysis, 7 of them have previously been linked to mitochondrial processes and are predicted to contain mitochondria targeting sequences. Within these, five genes are thought to contain RNA binding domains (*TBRG4*, and *MTPAP* as before, as well as *MRPP3, PNPT1* and *LRPPRC*). For genes not yet linked to mitochondrial processes, *SLC7A6OS* and *TGM3* show partial evidence of being targeted to mitochondria in some databases (*SLC7A6OS* is predicted to contain a mitochondria-targeting sequencing by IPSort, and both genes have a TargetP score >0.6 (Smith and Robinson, 2016)). Furthermore, we have previously described potential experimental links of some of these genes to mitochondrial processes. For example, *ZFP90* transgenic mice show altered expression of genes involved in oxidative phosphorylation and fatty acid elongation in mitochondria compared to wild type littermates (Yang et al., 2009), pointing to a potential role in mitochondrial processes.

In line with this new work, we have now added a description of the two new analyses to the Materials and methods subsection “Functional Annotation and links to Complex Disease”, as well as to the Results subsection " Functional characterization”. We also supplement existing annotation of the nuclear genes that we link to mitochondrial encoded gene expression using information obtained from Mitominer and associated databases.

2) Replication rate overall is low, which merits exploration. Is this due to power, technical variability, biological variability, or a higher false discovery rate than originally estimated? It is necessary to show analysis attempting to disentangle this through power calculation, simulation, and any relevant demonstration of biological or technical confounders. Could it be driven by differences in mt copy number, or cell state (leading to differences in mt-liked metabolic activity)?Given low replication rates and other concerns, it is noted that FDR may be susceptible to artifacts, and in general it is necessary to demonstrate that the FDR is well calibrated. The suggested method is to use permutation analysis (permuting a trait, leaving linkage disequilibrium intact) to establish an empirical null. In addition, please include raw p-values along with FDR for detected associations.

We have now performed power calculations to test whether a lack of replication in some cases may be a consequence of small sample sizes for replication datasets. To do this for each significant association, we obtained the correlation coefficient (r^2^) between the genetic variant and the expression of the associated mitochondrial encoded gene in the relevant discovery dataset (or largest dataset where the genetic variant is present, if multiple datasets are available for the tissue). We then used a power calculator (Purcell et al., 2003), specifying our estimate for the variance explained by the genetic variant (r^2^), the minor allele frequency, replication sample size and the significance threshold (0.05/61) in each case. Following this, we summed power values across all 61 associations and found that we have power to replicate ~40.5 of the 61 associations identified in the discovery phase. Beyond this, in terms of other biological/technical features, although we cannot measure mitochondrial copy number directly, we find significant differences between discovery and replication datasets for the proportion of mapped reads aligning to the mitochondrial genome in whole blood and subcutaneous adipose (Wilcoxon tests, P<0.05 after correcting for multiple tests). It is unclear whether this would influence our ability to replicate associations in these cases. We note that PEER factors (which we include as covariates in our association analyses) have been shown to correlate with known technical and biological features of RNA sequencing data (Stegle et al., 2010; Consortium et al., 2017; Glastonbury et al., 2018) and as such should control for some systematic variation across individuals. This additional analysis and discussion of caveats affecting the replication rate have now been included in the results section, as follows:

“In order to uncover potential reasons for a lack of replication for some associations, we performed power analysis using the variance explained by each genetic variant on the associated mitochondrial encoded gene expression level in the discovery dataset, together with the replication sample size, and find that ~40.5 associations would be expected to replicate (at P=0.00082). […] Even so, given the unexplained lack of replication in some cases, it is possible that false positives may contribute to our results.”

The methods for this analysis are described as follows:

“To perform power calculations, we obtained the correlation coefficient (r^2^) between the genetic variant and the expression of the associated mitochondrial encoded gene in the relevant discovery dataset (or largest dataset where the genetic variant is present, if multiple datasets are available for the tissue). We then used a power calculator (Purcell et al., 2003), specifying our estimate for the variance explained by the genetic variant (r^2^), the minor allele frequency, replication sample size and the significance threshold (0.05/61) in each case. Following this, we summed power values across all 61 associations.”

We have now also performed permutation analysis as suggested by permuting each trait and leaving linkage disequilibrium intact, in order to test whether the original association tests and FDR correction are well calibrated. We performed the following three types of permutation and correction:

1) Point-wise permutations: For each association that we originally identify as being significant at FDR 5% (64 variant-gene pairs), we performed 100,000 point-wise permutations for the relevant tissue type, mitochondria-encoded gene and nuclear genetic variant by randomly shuffling phenotypes. In each case, we then collected the test statistic across all 100,000 permutations to generate a null distribution, and then compared our observed test statistic against this to calculate an empirical P-value. For tissue types with multiple datasets (Whole Blood and LCLs) we performed permutations per dataset, combined these within a meta-analysis, and then derived the null distribution from the meta-analysis results. In each case, we also then followed the approach outlined in Ongen et al., 2016 to calculate a more precise P-value by estimating the underlying beta distribution of the null via maximum likelihood (using the “ebeta” function within the R package “EnvStats”). P-values obtained via this method closely follow raw P-values, and are now reported in Supplementary file 1 and Table 2.

2) Family wise error rate at the gene level: Following the approach above, we also performed 200 random permutations within each tissue per mitochondria- encoded gene for all nuclear genetic variants in each dataset. We then calculated the null distribution by selecting the largest test statistic across all nuclear genetic variants within each gene per permutation, and compared the observed test statistic for the relevant originally detected significant association to this null distribution in order to calculate the family wise error rate on the gene level. We also generated a more precise P-value in each case by estimating the underlying beta distribution of the null as above.

3) Family wise error rate across genes: Finally, we also repeated the above approach, but generated the null distribution in each case using the largest test statistic across all nuclear genetic variants and all 15 mitochondria-encoded genes per permutation to calculate the overall family-wise error rate. Again, we also generate a more precise P-value in each case by estimating the underlying beta distribution of the null.

We continue to present downstream analysis of all associations detected at FDR 5%, and clearly document which variants are significant through permutation analyses in Table 2 and Supplementary file 1, so that the reader may draw their own conclusions about the biological implications of variants detected by applying different significance thresholds. Results from this analysis are now noted in the second paragraph of the Results subsection “Nuclear control of mitochondrial gene expression”, and described in the Materials and methods section as follows: “To calculate P-values via permutation analysis, for each association that we originally identified as being significant at FDR 5% (64 variant-gene pairs), we performed 100,000 point-wise permutations for the relevant tissue type, mitochondria-encoded gene and nuclear genetic variant by randomly shuffling phenotypes. […] P values generated across all methods are shown in supplementary file 1.”

We also note that the direction of effect was previously incorrectly noted for one association in the replication data (rs62165226, *MTCO3* in LCLs). We have now corrected this in Table 2 and as a result, we find that 13 of the 61 associations significant at 5% FDR now replicate in independent tissue matched datasets (~21%). We have corrected this value in the Abstract, in the results section and in the Discussion.

3) In order to rule out false positive trans associations due to alignment error, it is necessary to evaluate whether there is sequence similarity between the candidate mt eQTL target genes and the nuclear regions of the genome, particularly for regions near the candidate eQTL variants. If there is sequence similarity, these could represent cis-eQTLs or nuclear eQTLs for genes that simply have some reads mis-mapping to mt. This is a similar problem to probe cross-hybridization observed for microarrays, and has been observed for RNA-seq false positive trans-eQTLs in GTEx and other studies.

To rule out false positive trans-genome associations driven by alignment error, we tested for the presence of nuclear encoded mitochondrial sequences (NUMTs) in the regions around each peak nuclear genetic variant that is associated with mitochondrial-encoded gene expression. NUMTs are mitochondrial DNA sequences that have transposed into the nuclear genome over evolutionary time scales, and as such often retain moderate to high sequence similarity with the mitochondrial genome. Specifically, we first obtained known NUMT sequences from the UCSC genome browser, which were generated by Simone et al., 2011, who used blastN to map nuclear chromosomes to the mitochondrial genome, setting the e-value threshold to 0.001. Sequences in this database range from 31 to 14904bp in length, with a similarity percentage ranging between 63% and 100%, thus the approach has the potential to tolerate a large number of mismatches between nuclear and mitochondrial sequences. Then, for each peak nuclear genetic variant that is associated with the expression of a mitochondrial-encoded gene we tested for the presence of at least 50bp of that gene in each NUMT (the shortest read length in our analysis) within 1MB of the corresponding nuclear variant. In total we find two such occurrences (out of 64 peak variants) and we observe ~4 and ~15 mismatches per 100bp in these sequences compared to the corresponding mitochondrial encoded sequence. Additionally, for each peak nuclear genetic variant that is associated with the expression of a mitochondrial-encoded gene, we also tested whether any 50bp segment of the mitochondrial-encoded gene also mapped to a nuclear gene (following the approach defined in (Saha and Battle, 2018)) that has its transcription start site within 1MB of the corresponding peak nuclear variant; we find no such occurrences. As such, it seems unlikely that alignment error is driving our associations.

We have now included these analyses in the results section as follows:

“To ensure that trans-genome eQTLs are not driven by alignment errors that are a consequence of sequence similarity between the nuclear and the mitochondrial genomes, we tested for the presence of nuclear mitochondrial DNA segments (NUMTs) in the regions surrounding each peak nuclear genetic variant. […] As such, alignment errors are unlikely to be driving the detection of trans-genome eQTLs for mitochondrial encoded gene expression.”

We also detail our approach in the Materials and methods section as follows:

“NUMT sequences were obtained from the UCSC genome browser track named ‘numtS’, and were generated by Simone et al., 2011, who used blastN to map nuclear chromosomes to the mitochondrial genome, setting the e-value threshold to 0.001. […] For each nuclear genetic variant associated with a mitochondrial encoded gene, we then tested whether any of the 50bp k-mers from the mitochondrial encoded gene aligned within a nuclear gene whose transcription start site fell within 1MB of the corresponding nuclear genetic variant.”

4) It is necessary to address questions regarding ancestry and linkage disequilibrium, particularly considering diverse ancestries represented in the RNA-seq data and potential differences with those in the GWAS data used. The manuscript evaluates SNPs in NHGRI GWAS catalogue in strong LD with the SNPs that control mtDNA gene expression. What LD reference is used? Is it the European LD reference calculated from 1000 genomes, or is it the in-sample LD from the datasets with RNAseq data? What were the populations (ancestries) in which the GWAS was performed? In other words, is the LD between SNPs calibrated for the GWAS or for the eQTL analyses? While it may not be possible to match LD perfectly between GWAS and eQTL analyses, these potential issues should be mentioned and some attempt made to determine if they are problematic. If the individual data is available, it is necessary to discuss single/per population analysis to show that their results are real and not artifacts of population structure and also likely consistent across populations. This would involve simply sectioning out the samples from each single population, and rerunning their analysis. The associations are not expected to be significant in the single population analysis since they'll have much smaller sample sizes, but the effect sizes and directions should be correlated. If this is not possible, then caveats of population structure need to be mentioned and the claims toned down accordingly.

Linkage disequilibrium values were calculated within the samples used to detect mitochondrial trans-genome eQTLs. This is now made clearer in the Materials and methods section:

“In order to identify whether genetic variants associated with mitochondrial encoded gene expression may play a role in complex disease, we first identified any SNP in linkage disequilibrium (r^2^>0.8, calculated using datasets and samples used in this study) with peak eQTL SNPs in any of the datasets used for the tissue type in which the association was identified.”

For GWAS variants that are in LD with variants associated with mitochondrial encoded gene expression, GWAS were conducted using individuals of Finnish (systolic blood pressure), European (diastolic – newly described) and European (Vitiligo) descent, whereas our eQTL analysis included individuals from diverse ancestries (although mostly of European descent). In order to try to match LD patterns more closely between our analyses and those conducted within GWAS, we repeated our analyses for the relevant genetic variants in whole blood data using only individuals of European descent. To do this we plotted the first two genetic principal components from samples in our study against those derived from 1000 genomes samples with known ancestry for any dataset that had associated RNA sequencing data from whole blood. We then selected samples that clustered with Europeans in 1000 genomes data by visual inspection and re-ran association analyses as before for whole blood data from CARTaGENE, TwinsUK and GTEx, before performing meta-analysis to calculate P-values. In all cases we find that the associations involving genetic variants in LD with GWAS variants remain highly significant. We also note that overlapping mitochondrial-encoded eQTL and GWAS variants are in strong LD in combined European populations surveyed by the 1000 Genomes project (r^2^>0.8 in all cases). These results imply that genetic variants associated with mitochondrial encoded gene expression are genuinely in LD with GWAS signals, however we agree that some caution should still be applied if populations within Europe are likely to generate further substructure in the data, which we have limited power to disentangle here. We now document this new analysis, together with a discussion of further population substructure, in the results section as follows:

“Genome-wide association studies considering blood pressure were conducted in individuals of Finnish (Kristiansson et al., 2012) and European descent (Ehret et al., 2016), and the study of Vitiligo was also conducted using individuals of European descent (Jin et al., 2016). […] These results imply that genetic variants associated with mitochondrial encoded gene expression are genuinely in LD with GWAS signals, however some caution should still be applied if populations within Europe are likely to generate further substructure in the data, which we have limited power to disentangle here.”

As well as in the Materials and methods section as follows:

“To test whether associations between nuclear genetic variants and mitochondrial encoded gene expression that overlap GWAS signals are significant in individuals of European descent, we plotted the first two genetic principal components against those derived from 1000 genomes samples with known ancestry for any dataset that had associated RNA sequencing data from whole blood. We then selected samples that clustered with Europeans in 1000 genomes data by visual inspection and re-ran association analyses as before for whole blood data from CARTaGENE, TwinsUK and GTEx, before performing meta-analysis to calculate P-values.”